# Conceptual Models and Calibration Performance—Investigating Catchment Bias

**Alexander J. V. Buzacott** * , **Bruce Tran, Floris F. van Ogtrop** and **R. Willem Vervoort**

Sydney Institute of Agriculture, School of Life and Environmental Sciences, The University of Sydney, Sydney 2006, Australia; bruce.tran1@gmail.com (B.T.); floris.vanogtrop@sydney.edu.au (F.F.v.O.); willem.vervoort@sydney.edu.au (R.W.V.)
* Correspondence: abuz5257@uni.sydney.edu.au; Tel.: +61-2-8627-1148

**Abstract:** Many lumped rainfall-runoff models are available but no single model can account for the uniqueness and variability of all catchments. While there has been progress in developing frameworks for optimal model selection, the process currently selects a range of model structures a priori rather than starting from the hydrological data and processes. In addition, studies on differential split sample tests (DSSTs) have focused on objective function definitions and calibration approaches. In this study, seven hydrological signatures and 12 catchment characteristics from 108 catchments around Australia were extracted for two 7-year time periods: (1) wet and (2) dry. The data was modelled using the GR4J, HBV and SIMHYD models using three objective functions to explore the relationship between model performance, catchment features and identified parameters. The hypothesis is that the hydrological signatures and catchment characteristics reflect catchment behaviour, and that certain signatures and characteristics are associated with better calibration performance. The results show that a greater percentage of catchments achieved a better calibration performance in the wet period compared to the dry period and that better calibration performance is associated with catchments that have greater cumulative flow and a steeper flow duration curve. The findings are consistent across the three models and three objective functions, suggesting that there is a bias in the studied models to wetter catchments. This study echoes the need to develop a conceptual model that can accommodate a wide variety of catchments and climates and provides a foundation to optimise and improve model selection in catchments based on their unique characteristics.

**Keywords:** calibration; conceptual model; catchment hydrology; differential split sample test

## 1. Introduction

Models are a necessary tool for the understanding and prediction of hydrology. They have a wide range of applications, from water resource management, hydro-engineering, flood and drought prediction, eco-hydrology and climatic studies [1–5]. In particular, they have the potential to aid our preparation and planning under an uncertain climatic future [6–8].

Simple (lumped) conceptual models are popular in hydrology. This popularity is likely due to simple models regularly performing as well or better than more complex models [9–12]. Conceptual models tend to have fixed structures with few parameters that represent specific hydrological features or processes and have minimal data requirements. A typical approach to catchment modelling is the selection of a model, followed by calibration of the model parameters using a particular objective function [13]. The calibration component is fundamental for parameter estimation and is used to assess the performance in relation to prediction of future values or scenarios [14]. The general aim of calibration is to define a set of reasonable parameters that capture the dominant behaviour of a

catchment for a given period of interest to provide a good 'fit' of the data. This should ensure that the model components and parameters retain their original physical meaning or interpretation [15,16].

While some catchment and model pairings can achieve good calibration performance, there are always some catchments that do not perform well relative to other catchments for example, References [17] (Figure 12), [18] (Figure 6). Even if good calibration performance is obtained, models frequently fail when tested in a new period outside which it has been trained. This suggests that the calibration approach does not always capture the actual processes that control the hydrological response within catchments [19]. The failure is often due to parameter equifinality [20] but can also be caused by model structural problems [21] or a scarcity of data [22]. In addition, the choice of objective function plays an important role in the success of model calibration [23] and obtaining appropriate representation of catchment behaviour [24]. For example, the Nash-Sutcliffe Efficiency (NSE) is sensitive to peak flows due to the use of squared deviations, making it poorly suited to low flow simulation [25] and may result in poor performance. There is also the effect of climate on model calibrations, where parameters and model performance vary substantially with different climate conditions [18,26,27].

Several studies in Australia have investigated the impact of variations in climate on model calibrations [8,28–31]. The research has mainly focused on improving calibration strategies to perform well in the traditional Differential Split Sample Test (DSST) [32,33]. This resulted in a framework [34] that identified, using a Pareto front approach, whether different objective functions can accommodate the non-linearity in the rainfall-runoff response [31] or whether data checks or changes in model structure are needed. However, the earlier research highlights that most of the models tested struggled with DSSTs [8] and even improved objective functions did not completely resolve this problem [31]. Elsewhere, there have been model comparison studies using a DSST design to evaluate model performance under contrasting climate conditions in Canada and Germany [7], France [35] and Ireland [36]. These studies highlighted the difficulty in identifying optimal model structures and parameters, with the solutions being either model ensemble runs [7,36] or flexible model structures [35].

As part of this research, there has only been limited emphasis on the catchment characteristics. The work of Saft et al. [30] showed that shifts in the annual rainfall-runoff relationship of catchments were mostly influenced by catchment characteristics that relate to pre-drought climate and groundwater and soil storage dynamics. Fowler [31] compared model performance with variables such as forest cover and mean slope but found little relationship between them and calibration performance. In addition, their work found little evidence of a difference between model structures on the model performance [31] (Figure 3). In other research, Trancoso et al. [21] link flow and catchment characteristics but they focused on the classification of hydrological behaviour viz a viz the Budyko and Dunne approaches and did not link this to model behaviour. Broderick et al. [36] used the Base Flow Index (BFI) as a way to distinguish catchments, as it is controlled by characteristics such as geology, vegetation, climate history [37]. They reported that catchments with a high BFI and low dynamic flow are best represented by more linear models, while a low BFI suggests a higher variability in runoff and requires a model with more parameters. What specific characteristics contribute to a better objective function score using the BFI, however, was not determined. Catchment area, the aridity index and an index that discriminates between surface water and groundwater dominated catchments (the ratio of seasonal runoff coefficients in summer and winter) was adopted by Esse et al. [35] to investigate a link between model structures and catchment characteristics. A priori models were developed based on sub-setting the catchments using the previously mentioned characteristics but they found that the pre-defined models were less consistent than either a fixed or flexible model and only useful for some types of catchments.

As a result, studies so far have not thoroughly attempted to investigate whether the calibration performance of models are dominated by certain catchment characteristics, which may reveal why models tend to perform poorly with certain catchments or transpose poorly between different climatic conditions and regions. The objective of this paper therefore is to approach the problem from a different

angle, investigating whether it is possible to identify specific catchment characteristics and hydrological signatures that result in better calibration performance for three popular conceptual rainfall-runoff models. To take into account the impact of differences in time periods for calibration [8,30], separate wet and dry periods are compared to assess the impact of non-linearity on the methodology and whether conclusions can be drawn about the generality of hydrological signatures, catchment characteristics and model structures on calibration performance. In addition, three different objective functions are tested to understand the variation due to this factor.

## 2. Materials and Methods

### 2.1. Study Area and Data Sources

For this study a range of catchments around Australia was identified, located in areas with varying geologies, topographies, climates and climatic regions, vegetation and land use (Figure 1). Streamflow data was obtained from the Australian Bureau of Meteorology (BoM) Hydrologic Reference Station (HRS) project (http://bom.gov.au/water/hrs/about.shtml). A detailed overview of the HRS project is provided in Zhang [38]. There is no missing data in the record as the BoM gap fills the data using the Génie Rural à 4 paramètres Journalier (GR4J) model [39] (http://bom.gov.au/water/hrs/methodology.shtml). However, as this study also uses the GR4J model (as indicated later), we will discuss possible implications.

The time period is split into: (1) wet period 1990–1996 and (2) dry period 2000–2006 to investigate for an effect on model performance. A seven year period was chosen to incorporate various high and low flow events into the data record. Annual rainfall anomalies were used to determine the distinct wet and dry periods. The Millennium Drought [40] that occurred in south eastern Australia was deliberately chosen as the dry period as the majority of the study catchments are located in this region. However, northern Australia and western Australia did not experience similar dry periods and we will discuss how this might have affected our results.

The HRS used in this study were selected on the basis of: (1) Streamflow was available for both time periods and (2) over 90% of the streamflow record had the highest quality code. This resulted in many catchments from the Northern Territory, Tasmania and central to northern New South Wales being excluded from the final set. Due to the large size of one catchment (A0020101, 119,034.0 km$^2$) it was also excluded from the final set, with the next largest being 2225.6 km$^2$. This resulted in a set of 108 catchments which were used for the analysis (Figure 1).

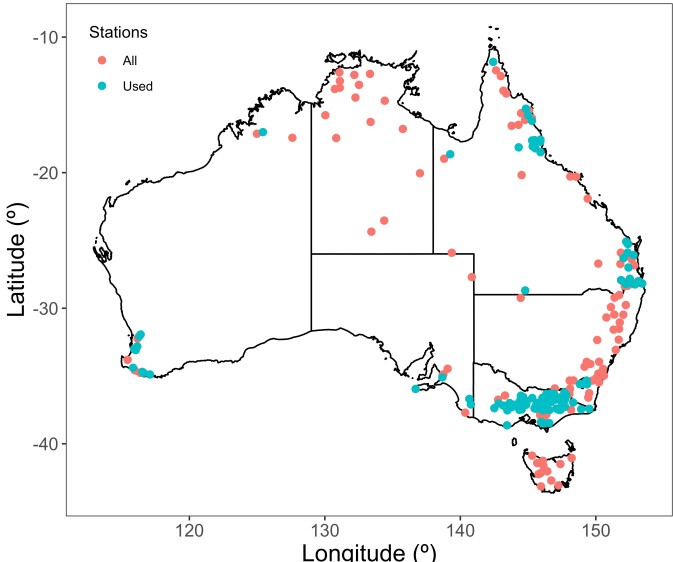

**Figure 1.** Map of all the Hydrological Reference Stations available in Australia and those selected for this study. Stations were selected if the streamflow record had over 90% of the highest quality code.

Catchment daily rainfall, maximum temperature, minimum temperature and Morton's potential evapotranspiration [41] were extracted from the SILO data drill database [42] (http://silo.longpaddock.qld.gov.au). Maximum and minimum temperature were averaged for an estimate of daily mean temperature.

### 2.2. Model Selection

Three conceptual lumped models were selected for analysis: GR4J, SIMHYD [43] and Hydrologiska Byråns Vattenbalansavdelning (HBV) [44]. These models were selected based on their extensive use in water resources research and differences in their structural complexity. Table A1 provides a summary of the model structure and parameter ranges. Wide parameter ranges were selected for each model to cover the variability in catchments. Each model run has an initialisation period of 100 days for which the results are discarded.

The *R* package HYDROMAD [45] was used as the modelling framework in this study. The version of SIMHYD used is based on the description in Chiew et al. [17]. Many versions of the HBV model exist. The model code from a version of HBV as used in Reference [46] was adapted for use with HYDROMAD in this study.

### 2.3. Objective Functions

The Kling-Gupta Efficiency (KGE) [47], Nash-Sutcliffe Efficiency (NSE) [48] and R-Squared with log transformed data ($R^2_{log}$) were selected as objective functions and performance metrics for this study. The KGE and NSE are the most frequently used objective functions in hydrology. $R^2_{log}$ is adopted as transformation of the timeseries reduces the leverage of high flows and increases the impact of mid to low flows [49]. This is often beneficial for parameter selection for drier conditions. Parameter optimisations were performed using the Shuffled Complex Evolution—University of Arizona (SCE-UA) algorithm [50]. This algorithm routine performs well with a relatively small amount of free parameters [51,52]. The number of complexes was set as the number of free model parameters plus one, the relative improvement factor was set to $10^{-5}$ and the maximum number of iterations was set to 10,000. The SCE-UA optimiser is applied for each model, catchment and objective function combination, giving a total of 972 optimisations.

The objective functions are defined as follows:

$$KGE = 1 - \sqrt{(r-1)^2 + (\alpha - 1)^2 + (\beta - 1)^2}, \tag{1}$$

where *r* is the linear correlation between simulated discharge ($Q_{sim}$) and observed discharge ($Q_{obs}$) and $\alpha$ and $\beta$ are the ratio of the standard deviation and mean for $Q_{sim}$ and $Q_{obs}$, respectively.

$$NSE = 1 - \frac{\sum_{t=1}^{t=T}(Q_{sim}(t) - Q_{obs}(t))^2}{\sum_{t=1}^{t=T}(Q_{obs}(t) - \overline{Q_{obs}})^2}, \tag{2}$$

where *T* is the total number of time steps, $Q_{sim}(t)$ and $Q_{obs}(t)$ the respective values of simulated and observed discharge at time *t* and $\overline{Q_{obs}}$ the mean observed discharge.

$$R^2_{log} = 1 - \frac{\sum_{t=1}^{t=T}(\log(Q_{sim}(t) + \epsilon) - \log(Q_{obs}(t) + \epsilon))^2}{\sum_{t=1}^{t=T}(\log(Q_{obs}(t) + \epsilon) - \log(\overline{Q_{obs}} + \epsilon))^2}, \tag{3}$$

where $\epsilon$ is the lowest decile of the non-zero values of *Q*.

The Akaike Information Criterion (AIC) [53] is used as a standardised score to compare the performance of the models and objective functions. The AIC penalises the Root Mean Square Error (RMSE) for the number of the parameters in the model [49]:

$$AIC = m \log_n(RMSE) + 2p, \tag{4}$$

where *m* is the number of points used in calibration and *p* is the number of free parameters used in the model. The RMSE is expressed as:

$$RMSE = \sqrt{\frac{1}{T}\sum_{t=1}^{t=T}(Q_{sim} - Q_{obs})^2}. \tag{5}$$

*2.4. Analysis*

Seven hydrological signatures were calculated from the streamflow data and 12 physical characteristics were identified and are summarised in Table 1. The hydrological signatures used are—low flow (FDC.low), mid flow (FDC.mid), high flow (FDC.high), autocorrelation (AC), peak distribution (peaks), FDC.slope, cumulative flow (CF). They are based on the studies in References [54,55]. FDC.low, FDC.mid, FDC.high are single quantities reflecting flow at the 90th, 50th and 1st quantiles, respectively, from the flow duration curve of observed data. AC is the lag-1 autocorrelation coefficient, which is the correlation between two daily flows of the hydrograph and useful for understanding the influence of memory in the system [55]:

$$AC = \frac{\sum_{t=1}^{t=T}(Q(t) - \overline{Q})(Q(t+1) - \overline{Q})}{\sum_{t=1}^{t=T}(Q(t) - \overline{Q})^2}, \tag{6}$$

where $Q(t)$ is the value of discharge at time $t$ and $\overline{Q}$ the mean value of discharge.

The peaks variable is a measure of the differences in the peak heights in the flow record and is calculated as the slope of the 10th to 50th percentile in the flow duration curve [55], in other words, a greater number reflects more variable flow, while a smaller number reflects more even flow. The variable FDC.slope is a similar measure but calculated as the slope of the flow duration curve between the 25th and 75th percentile.

**Table 1.** Descriptions of the hydrological signatures and catchment characteristics used in this study.

| | Unit | Description |
|---|---|---|
| *Signature* | | |
| FDC.low | mm | 90th quantile of flow |
| FDC.mid | mm | 50th quantile of flow |
| FDC.high | mm | 1st quantile of flow |
| FDC.slope | - | Slope of the middle part of the flow duration curve |
| AC | - | Correlation coefficient between 2 points-1 day |
| Peaks | - | Calculates difference between the height of peak events |
| CF | mm | Cumulative flow over the time period |
| *Characteristic* | | |
| Qcv | - | Coefficient of variation of annual streamflow |
| P/PET | - | Aridity index: ratio of annual P to annual PET |
| Area | km$^2$ | Catchment area |
| Elevation | m | Mean elevation within the catchment |
| Elevation range | m | Elevation range within the catchment |
| PAWC | mm | Mean plant available water capacity in the top 1 m of soil |
| Slope | ° | Mean slope within the catchment |
| Soil depth | m | Mean depth of soil within the catchment |
| Stream length | km | Sum of the length of streams |
| Stream density | km/km$^2$ | Density of streams (stream length/catchment area) |
| Woody cover | % | Mean percent cover of woody vegetation |
| River type | - | Perennial: no flow $\leq$ 1% of the time |
| | | Ephemeral: no flow $>$ 1% of the time |

The physical characteristics selected here are based on the studies by Vaze et al. [28] and Saft et al. [30]. The annual streamflow coefficient of variation (Qcv) and an aridity index, the ratio of

annual precipitation to annual potential evapotranspiration (P/PET), are two characteristics calculated to explore the effect of long term variability and the catchment dryness. Qcv is calculated using a 29 year period where there was HRS data available for all catchments (1983–2011). This same period was used to calculate the P/PET characteristic using SILO data. Catchment area is determined from delineated catchment boundaries extracted from the BoM Geofabric V2.1 product (http://www.bom.gov.au/water/geofabric/), a stream and nested catchment framework for Australia [56]. These catchment boundaries were used to calculate or extract many of the remaining characteristics. Mean elevation, elevation range and mean slope were determined from the processed 1-second (~30 m) Shuttle Radar Topography Mission (SRTM) digital elevation model data for Australia [57]. Mean soil depth was extracted from the Soil and Landscapes Grid of Australia [58] while plant available water capacity (PAWC) is from Australian Soil Resource Information System [59]. Stream length is the sum of the length of streams within the catchment boundary from the Geofabric stream network. Stream density is calculated as the stream length divided by the catchment area. Woody cover was extracted from the Australian Woody Vegetation Cover product [60] which is derived from 30 m Landsat data. River type is whether the stream is ephemeral ($n = 57$) or perennial ($n = 51$). A stream was classified as being ephemeral if there was zero flow more than one percent of the time and a perennial as having zero flow less than or equal to one percent of the time. The dry period (2000–2006) was used for stream type classification. A numerical summary of these characteristics can be found in Table 2.

**Table 2.** Numerical summary of the catchment characteristics by catchment type: ephemeral ($n = 57$) and perennial ($n = 51$).

| Characteristic | Ephemeral | | | Perennial | | |
|---|---|---|---|---|---|---|
| | Mean | Median | SD | Mean | Median | SD |
| Qcv | 17,921 | 7532 | 35,273 | 59,007 | 23,584 | 122,439 |
| Aridity index (P/PET) | 0.57 | 0.54 | 0.20 | 0.90 | 0.90 | 0.32 |
| Catchment area (km$^2$) | 1028 | 304 | 2907 | 727 | 316 | 1987 |
| Mean elevation (m) | 427 | 398 | 252 | 656 | 678 | 303 |
| Elevation range (m) | 575 | 491 | 340 | 918 | 990 | 434 |
| Mean PAWC (mm) | 93 | 88 | 31 | 125 | 127 | 38 |
| Mean slope (°) | 5.4 | 4.5 | 3.7 | 11.5 | 12.2 | 5.3 |
| Mean soil depth (m) | 0.97 | 0.96 | 0.10 | 0.99 | 1.00 | 0.13 |
| Stream length (km) | 689 | 232 | 1631 | 555 | 234 | 1435 |
| Stream density (km/km$^2$) | 0.76 | 0.79 | 0.21 | 0.81 | 0.82 | 0.15 |
| Woody cover (%) | 33 | 31 | 19 | 50 | 51 | 19 |

Principal component analysis (PCA) and the machine learning method Random Forest [61] are used to investigate the relationships between hydrological signatures, catchment characteristics and objective function values. A clear explanation of how PCA works is outlined in Euser et al. [55]. The PCA is performed using log transformed values for catchment area, slope, stream density and stream length and all values of all the hydrological signatures except for AC. They are also scaled due to the different units between the signatures and variable range in values between the catchment characteristics and the hydrological signatures.

Random Forest is used as it provides a natural way to explore variable importance in relation to the objective function scores. Variable importance is determined by calculating the mean squared error on the Out-Of-Bag (OOB) data (data that is excluded from the bootstrap subset) for each prediction tree and then the same is performed after random permutation of each predictor variable. The mean difference between the OOB error and post-permutation error over all trees is taken and normalised by the standard deviation of differences. If the variable is of little importance, the effect of random permutation on the OOB error is minimal. The *R* package *randomForests* [62] is used for all Random Forest model computations. The amount of trees ($n_{tree}$) was set to 501 to avoid equal votes, while the number of points per terminal node (nodesize) and the number of variables sampled at each split

($m_{try}$) were left at the default settings of 5 and the number of parameters divided by 3, respectively. The latter variable, $m_{try}$, is sensitive in Random Forest models [63], however the default method of setting the variable is often a good choice [62]. Other values were evaluated but there were minimal differences in the OOB error estimates. The predictors are left untransformed for the Random Forest models as the output is invariant to monotone transformations [64].

The results from the model calibrations for the wet and dry periods are merged before analysis by PCA and Random Forest models. A new variable, Period, is used with the Random Forest models to classify the calibration period.

## 3. Results

### 3.1. Overview of the Observed Signatures

A comparison of the hydrological signature values between wet and dry periods for the observed data is presented in Table 3. The wet period had higher observed values than the dry period across all signatures except for autocorrelation (AC), a demonstration of the impact the Millennium Drought had on the study catchments. All signatures have positively skewed distributions apart from AC. The lower values for the low and mid flow signatures indicate the variability of hydrological regimes in Australia. The signatures are moderately to strongly correlated to each other, except for AC which was weakly correlated with the other signatures. For example, the Pearson's correlation between AC and FDC.high (or high flows in the flow duration curve) was 0.01 in the wet period and 0.09 in the dry period. The correlation between AC and the other signatures ranged from 0.21 to 0.34 across both periods.

**Table 3.** Summary of hydrological signatures for the observed catchment flow data.

| Period | Statistic | CF | FDC.low | FDC.mid | FDC.high | FDC.slope | AC | Peaks |
|--------|-----------|------|---------|---------|----------|-----------|-------|-------|
| Wet | Median | 1183 | 0.018 | 0.121 | 5.683 | 0.223 | 0.723 | 2.110 |
| Wet | Mean | 1719 | 0.091 | 0.330 | 7.094 | 0.505 | 0.724 | 3.313 |
| Dry | Median | 739 | 0.005 | 0.065 | 2.793 | 0.149 | 0.763 | 1.438 |
| Dry | Mean | 1375 | 0.064 | 0.236 | 4.832 | 0.419 | 0.741 | 2.504 |

### 3.2. Model Performance

All of the catchment, model and objective function combinations successfully converged using the SCE-UA optimisation routine, resulting in $n = 972$ total optimised scores to analyse. Figure 2 shows a figure similar to Figure 12 in Chiew et al. [17], indicating the probability of exceeding an objective function value for each objective function and model. This demonstrates overall that the wet period has generally higher performance than the dry period for each objective function optimisation. This behaviour was consistent for each model and objective function pairing. This figure shows the GR4J and HBV models performing similarly for the KGE and NSE objective functions in the wet period, with GR4J performing slightly better. The same is observed in the dry period until 85% probability of exceedance, where there is a steep decline in objective function scores for GR4J and SIMHYD had the poorest performance of the models for these two objective functions in both simulation periods. For the $R^2_{log}$ optimisations the GR4J model clearly performed the best in the wet period but there is little difference in the results in the dry period. The wet and dry periods for HBV and SIMHYD have similar performance until approximately 60% probability, then there is a divergence where the dry period results in lower objective function values.

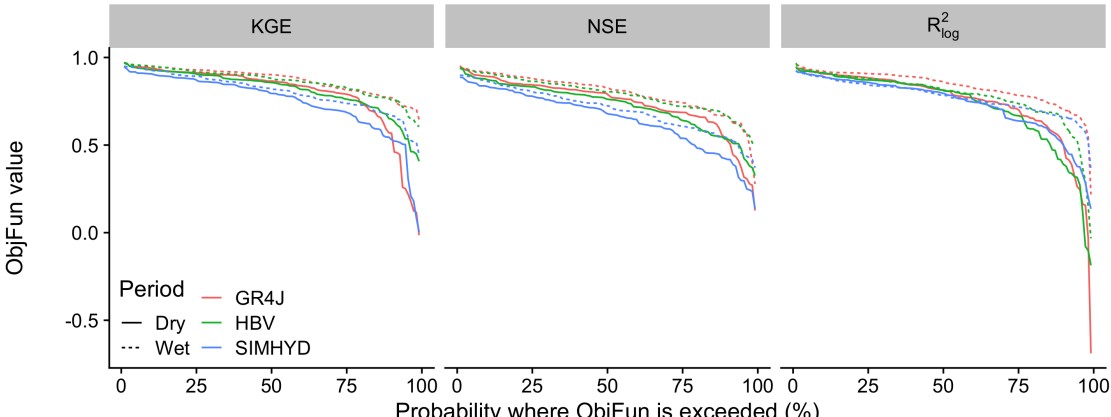

**Figure 2.** Probability of exceeding a specific objective function value across the catchments. Objective functions are in different panels and the models used for calibration are in different colours. The dry period is a solid line and the wet period is a dashed line.

Figure 3 is the same plot except that it uses the AIC instead of the objective function values as a performance indicator across models and objective functions. This shows the dry period achieving more negative (better) AIC scores and small differences in the performance of each model. The order of models performing the best across the three objective functions overall is: GR4J, HBV and then SIMHYD, as it is in Figure 2.

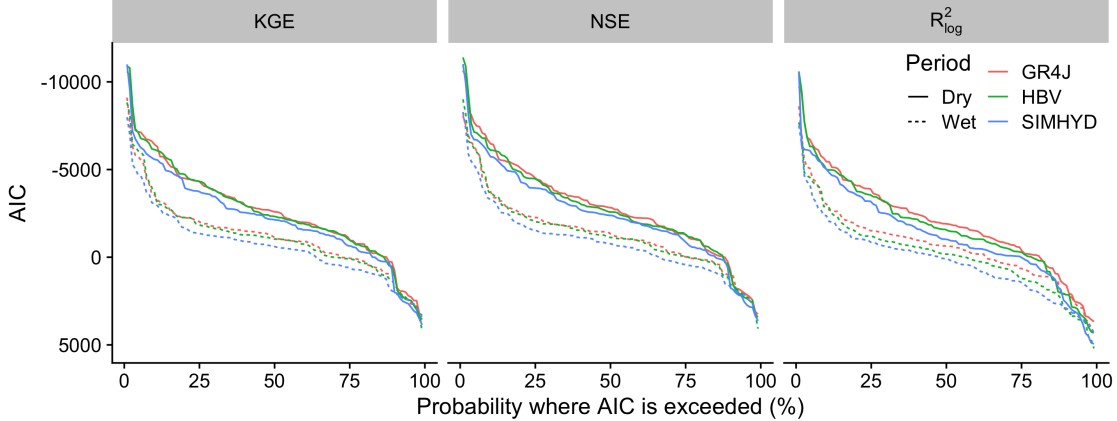

**Figure 3.** Comparing probability of exceeding a specific objective function value across the catchments standardised using the Akaike Information Criterion (AIC). Objective functions are in different panels and the models used for calibration are in different colours. The dry period is a solid line and the wet period is a dashed line.

### 3.3. Principal Component Analysis

In all cases the first principal component PC1 explains a high percentage of the overall variance in the data (45–50%), while PC2 explains a much smaller amount of variance (14–15%) (Figure 4). A closer inspection of the PCA plots indicates that PC1 is strongly related to the flow signatures, while PC2 explains catchment characteristics.

Across the model and objective function pairs, PC1 shows a clear separation between ephemeral and perennial streams. Catchment area and stream length have the same vector dimensions and direction for all cases and are the only variables in the direction of the ephemeral streams along PC1. These two variables slightly oppose the objective function vectors, with the only exception the HBV and and KGE combination. They are more directly opposed to soil depth and woody cover, indicating that larger catchments tend to have shallower soils and less woody vegetation cover.

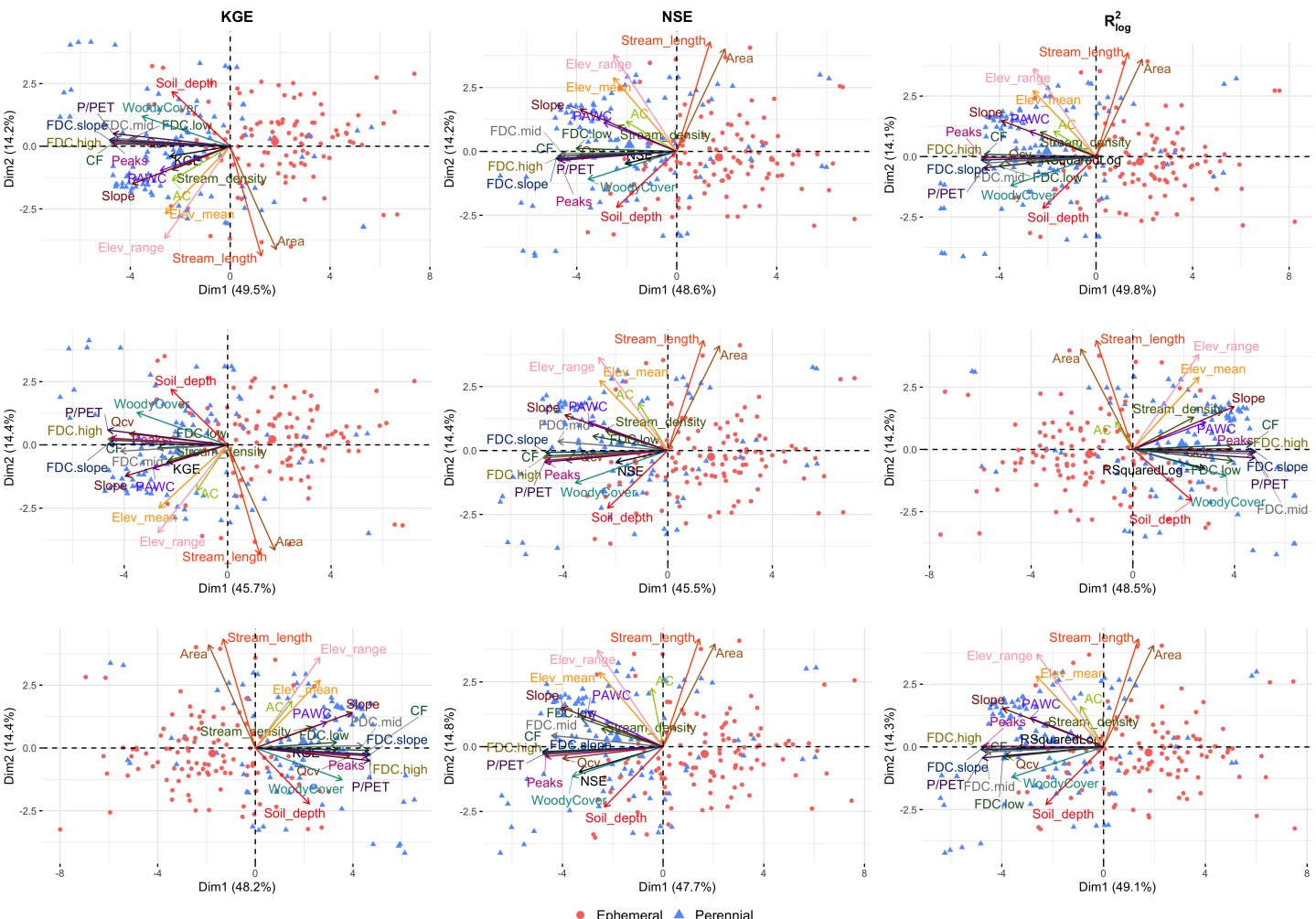

**Figure 4.** Biplots of the principal component analyses for each model and objective function. The models are in rows and the objective functions are in columns. The $R^2_{log}$ objective function vectors are labelled as RSquaredLog inside the biplots.

The remaining vectors are in the direction of the perennial streams. The flow duration curve signatures FDC.low, FDC.mid, FDC.high, as well as FDC.slope, Peaks and CF are all strongly pointing in the direction of perennial streams. This is a reflection of the high collinearity between these variables. AC was often orthogonal to these vectors but still in the direction of the perennial streams. The remaining physical characteristics are spread across the perennial stream dimension. Qcv and P/PET are closely related to the hydrological signatures, an indication that these signatures are related to catchments that are less arid and have high interannual stream variability. The two elevation parameters, elevation range and mean elevation and the AC vector were aligned for all plots except for the HBV and SIMHYD $R^2_{log}$ pairings. Here it was still in a similar direction, suggesting catchments with a large mean elevation and elevation range have higher stream autocorrelation. Woody cover and soil depth are closely aligned to each other but orthogonal to PAWC which is closely aligned with catchment slope.

In all cases the objective function vectors are in the direction of the perennial streams. They are in close proximity to the hydrological signatures and in some cases directly aligned with them. The GR4J model results for Kling-Gupta Efficiency (KGE) and Nash-Sutcliffe Efficiency (NSE) do not show any variables directly in line with the objective function but they are near the hydrological signatures, plus stream density, slope and PAWC for KGE and woody cover for NSE. The $R^2_{log}$ objective function is directly in line with FDC.low and P/PET for the GR4J model. The HBV model has PAWC, slope and stream density directly aligned with the KGE objective function, while the NSE objective function is between woody cover and a collection of the hydrological signatures and Qcv and P/PET. Woody cover and FDC.low are the two closest in proximity for the HBV and $R^2_{log}$ pairing. The SIMHYD model had FDC.slope directly aligned with the KGE and $R^2_{log}$ objective functions, with the latter very close to FDC.high and the Peaks signature. The NSE and SIMHYD pairing did not have any variables directly aligned. Woody cover and Qcv were the two closest variables.

The objective function vectors are not overwhelmingly strong but do suggest that there is a bias towards perennial streams with dynamic flow. Woody cover is regularly close to the objective function vectors. More heavily forested catchments have been shown to reduce overall yield but maintain flows [65].

### 3.4. Random Forest

The measures of variable importance for predicting the objective function value by the Random Forest models are in Figure 5 and the percentage of variance explained in Table 4. The amount of variance explained for SIMHYD is greater than GR4J and HBV. Models for the $R^2_{log}$ objective function had the highest levels of variance explained.

**Table 4.** The percent amount of variance explained by the Random Forest models when predicting the objective function and Akaike Information Criterion (AIC).

| Model | ObjFun Value | | | AIC | | |
|---|---|---|---|---|---|---|
| | **KGE** | **NSE** | $R^2_{log}$ | **KGE** | **NSE** | $R^2_{log}$ |
| GR4J | 37.82 | 40.21 | 57.86 | 84.86 | 86.48 | 76.17 |
| HBV | 47.20 | 45.35 | 57.13 | 87.76 | 88.55 | 78.54 |
| SIMHYD | 60.71 | 61.80 | 73.13 | 85.10 | 87.71 | 79.87 |

The hydrological signatures are of greater importance to explain the performance of the GR4J model (Figure 5). For the top 5 variables across the objective functions, only the NSE had physical variables such as soil depth and elevation range associated with this model. The HBV model shows a dominance of physical characteristics in the top 5 important variables for the KGE and NSE objective functions, while for $R^2_{log}$ these are not as important with soil depth the only physical characteristic in the top 5. The SIMHYD model shows greater consistency to important variables for the KGE and NSE

objective functions. They have the same variables in the top 5 (Qcv, PAWC, CF, AC and soil depth), with only CF and soil depth switching places. The results differ for $R^2_{log}$, where leaving out mostly flow duration curve variables generates lower performance.

By objective function, it is clear that the Peaks is important for predicting $R^2_{log}$ values and the NSE is more sensitive to physical characteristics (e.g., stream density, soil depth). The KGE had the most variation of variable importance and only contained CF in the top 5 variables for the three models.

The results show that there is some dependence on the model and objective function pairing, however there is a general pattern. The variable CF is frequently important to the objective function and model pairs. It is in the top 3 important variables for the KGE and NSE objective functions and top 5 for $R^2_{log}$ across all models. This suggests that catchments with higher cumulative flows achieve greater objective function scores.

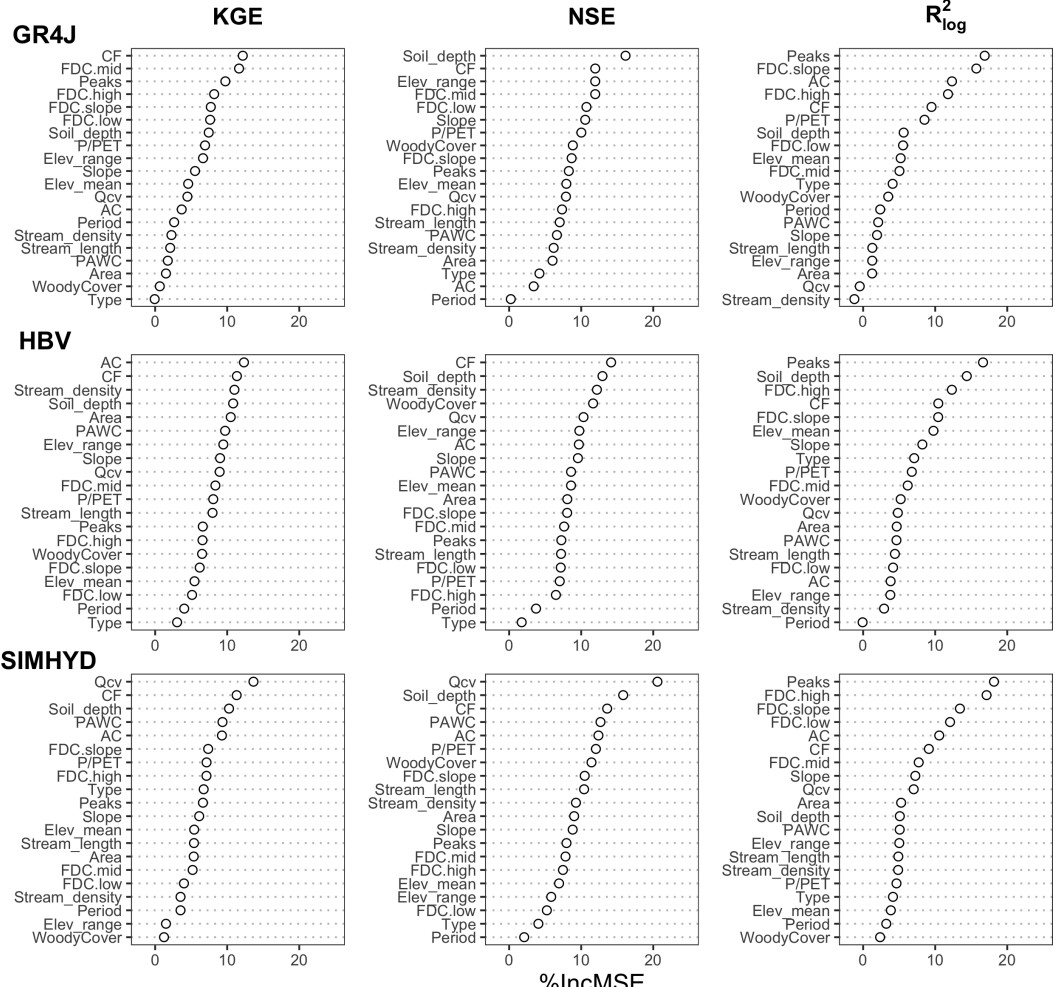

**Figure 5.** Variable importance in the Random Forest models as measured by the percent increase in mean squared error in the Out-Of-Bag data. The models are in rows and the objective functions are in columns.

Using the AIC, as a standardised score across the objective functions as the response with Random Forest models, produced more consistent results (Figure 6). The amount of variance explained with these models is greater (Table 4) and the amount of variation across models and objective functions is lower. The most important variables were Qcv, CF and AC. Given better AIC scores were found in the dry period (Figure 3), this means that lower values of interannual stream variation, lower cumulative flow and higher autocorrelation result in better scores. This is particularly clear for the KGE and NSE objective functions, where Qcv, CF and AC are frequently important variables. Qcv remains an

important variable for the $R^2_{log}$ simulations, while CF was still in the top five for each model. The flow period is frequently an important variable, highlighting the effect large residuals during the wet period have in relation to the AIC values. Stream density is in the top five variables in each case. Low density networks are typical of larger and drier catchments, where there is a less flashy hydrograph and generally lower flows that generate smaller residuals and accordingly have lower AIC values.

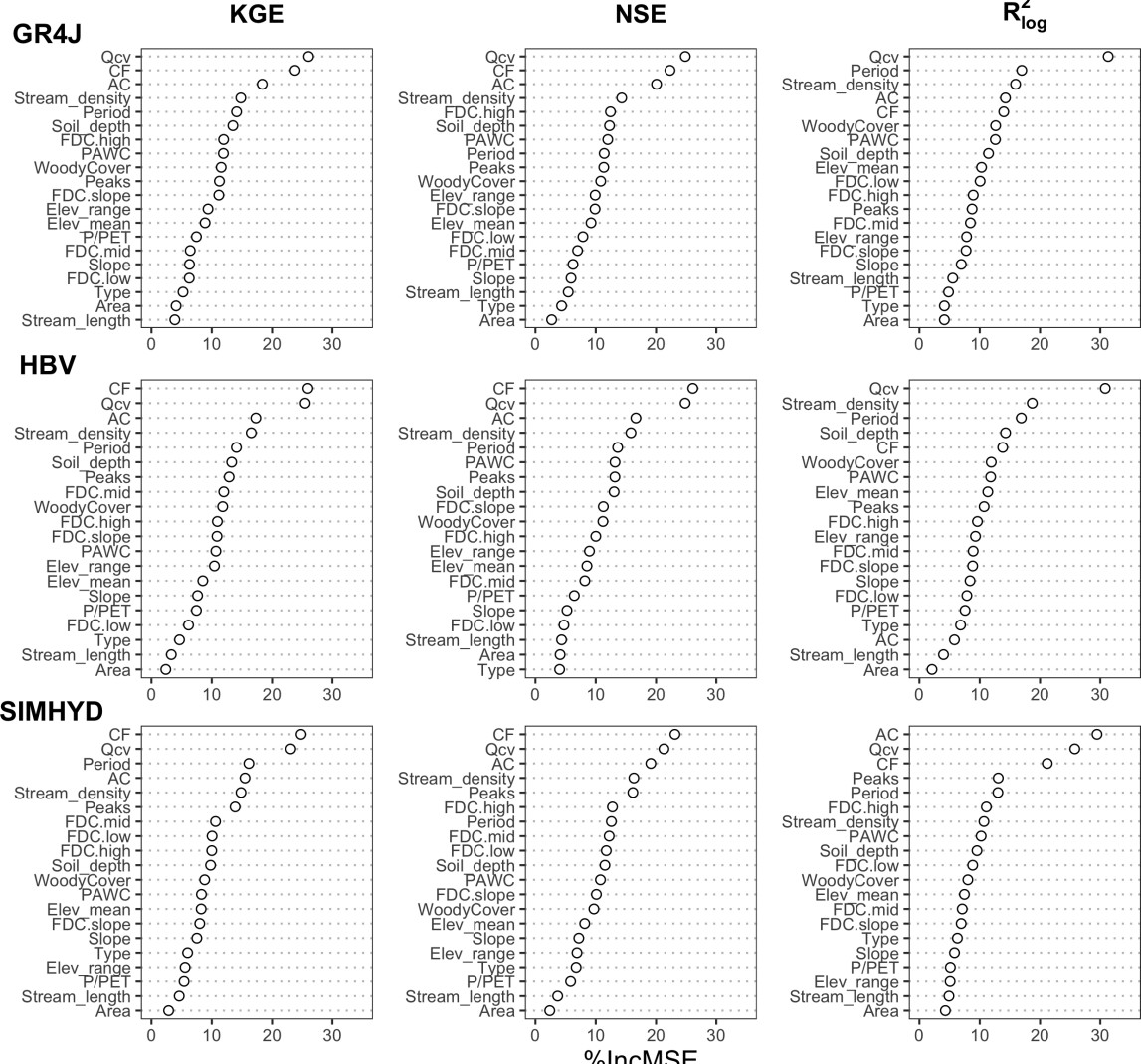

**Figure 6.** Variable importance in the Random Forest models, as measured by the percent increase in mean squared error in the Out-Of-Bag data, when predicting the Akaike Information Criterion. The models are in rows and the objective functions are in columns.

## 4. Discussion

### 4.1. General Performance

As part of understanding non-stationarity in models [19], a number of papers in Australia have concentrated on the DSST and reasons why this tends to fail (e.g., References [8,30,31]). This has suggested that causes for the non-stationarity might be due to changes in the rainfall-runoff relationship [30] and optimal parameter combinations can be identified using Pareto fronts that might perform better [8]. Furthermore, using different objective functions may identify parameter combinations that perform better in a DSST.

Here we take a slightly different approach, hypothesising that the failure of models in the DSST might simply be due to the fact that model structures are biased to wetter environments. Following

Serinaldi et al. [66], we hypothesise that aspects of high flow, low flow [21] and memory [67] would most explain the variation in flow. This extends the work by Euser et al. [55] and Saft et al. [30] linking hydrological signatures and catchment characteristics to structural consistency in models. This focuses on catchments that are towards the tail of the probability distributions in Figure 2 and the characteristics of the flow distributions in these catchments.

There was a clear difference between the probability distributions of objective function scores (Figure 2), which shows the calibrations in the wetter climatic period (1990–1996) achieving higher performance than in the the dry period (2000–2006). This is in line with other studies showing wetter catchments performing better [35]. The performance distribution using the AIC as a standardised score in Figure 3 surprisingly showed the dry period achieving lower AIC values, thus suggesting better performance during the dry period. There are also minimal differences between the three models using the AIC. The difference between the GR4J and HBV was smaller than the difference between the dry and wet periods, while this is not true for SIMHYD (Figure 2). This is in contrast to other studies, where Fowler [31] found more difference in the performance between models than between objective functions.

The results in Figure 2 highlights the issue of trying to make comparisons with an RMSE based method, which is sensitive to heteroscedasticity of residuals. Other studies have expressed this for the NSE, where they doubt the validity of using it for comparative purposes [68,69]. This points to a broader issues in terms of making comparisons across performance measures which are routinely done in hydrological studies. What is considered "better" for one performance statistic, can be worse under another performance statistic. For example, when maximising the NSE there tends to be an underestimation of flow variability, while there are also issues with bias due to scaling the observed values by the standard deviation [47]. The KGE partially corrects this by using the Euclidean distance of the three components of the decomposed NSE, but they are still both dominated by the linear correlation component. This makes comparisons across objective functions problematic and this is why this study concentrates on explaining what catchment characteristics determine the level of the performance in a calibration.

The KGE and NSE objective functions were selected in this study based on their popularity and the $R^2_{log}$ was selected as the transformation of the flow series makes it more sensitive to low flow deviations which suit many of the flow regimes in Australia. Issues have been raised with the use of least-square approaches calibration in drying climates and using modified objective functions, like the Split-KGE, will make calibrations more robust [31]. Using modified objective functions like the Split-KGE may reduce the difference between the models and objective functions or biases to particular climatic periods but the performance distributions in Fowler [31] were still long-tailed. Multi-objective function calibrations are another alternative [24], however the aim of this study was to examine standard modelling behaviour by hydrologists.

### 4.2. Interpretation of the PCA

The PCA biplots in Figure 4 support the hypothesis that better performance is related to wetter environments that have dynamic flow behaviour. More specifically, there is a strong separation of perennial and ephemeral rivers and the objective function vectors all point towards the perennial side, irrespective of the model. This is accompanied by close alignment of the objective function vectors with the flow signals (e.g., FDC.high, FDC.mid, FDC.slope, Peaks, CF) and the catchment characteristics Qcv, P/PET, stream density and woody cover that (in Australia) tend to be indicative of wetter environments. That Qcv is strongly associated with the perennial streams is not surprising. While we typically think of drier systems as being highly variable, wetter systems experience truly more variability and this is reflected in the Qcv and P/PET alignment.

Interestingly, using the PCA classification from Euser et al. [55] all the models are fairly consistent structurally. This is indicated by most of the arrows in the PCA plot pointing in the same direction.

This is despite a bias for catchments with more flow and variability. In contrast to Euser et al. [55], the complexity of the models did not seem to affect the consistency.

The PCA plots also indicate that the models have lower performance in larger catchment areas for these lumped conceptual models. This is in contrast to Esse et al. [35], who found that larger catchments performed better than smaller ones. The mean catchment size in this study is 911 km$^2$, somewhat larger than 566 km$^2$ as in Esse et al. [35] and possibly French catchments have lower climate and physical variability compared to the catchments used here. Larger catchments in Australia are typically found in the flatter plains west of the Great Dividing Range, experience reduced rainfall and greater evapotranspiration. This creates significant trends and correlation in some of the catchment characteristics in Australia.

### 4.3. Random Forest

We opted to use Random Forest instead of other methods, such as multilinear regression [30], because Random Forests can naturally provide a way to inspect variable importance, robustly handle a combination of continuous and categorical variables and has the ability to handle complex relationships between predictor and response variables. The model results provide a summary of the interaction of the model structure, the specifics of the objective function and the chosen physical and catchment characteristics.

The results here show some support for the hypothesis. Cumulative flow was a top ranking variable in model and objective function pairing, indicating that catchments with higher total flows achieve better performance. Surprisingly, the variables associated with describing catchments during the high flow period were the most important to the $R^2_{log}$ objective function and not the KGE or NSE. As this objective function is generally considered to fit low flow behaviour better, it was expected that FDC.low would be more important in explaining the objective function score. To understand this result and to assess the relationship in the Random Forest models, pairwise plots of the $R^2_{log}$ score and the different parts of the flow duration curve were inspected. This revealed that there is a strong positive correlation between FDC.high and FDC.mid, while there is a weaker positive correlation to FDC.low. In comparison to the NSE, which had the same strong positive correlation for FDC.high and FDC.mid, FDC.low had a long distribution of poor objective functions NSE values. In effect, the transformation of the $R^2_{log}$ helps fitting of the lower end of the curve and avoids poor performance but the upper end is still important for determining overall performance, most likely because of the larger residuals associated with higher flows. Additionally, despite clear separation of the stream type in the PCA biplots, the type variable did not rank highly in the Random Forest models. Additionally, the flow period was frequently a poorly ranked variable.

In this study the separation of dry and wet periods is less rigorous then in the earlier studies [8,30,31] and this may explain the lack of influence of dry and wet periods in the Random Forest models. Soil depth was an important variable for every model with the NSE objective function and appears to be important to the HBV model, yet there is a minimal difference of soil depth between the perennial and ephemeral streams (Table 2) and the distribution of soils and objective function scores is fairly uniform. Most likely soil depth is a proxy for overall catchment storage of water and factors such as "valley bottom flatness" that influence hydrology [30].

Using the AIC as the response variable yielded more consistent results with respect to the variables and the objective function and model pairings. Here, the results show that interannual variation, cumulative flow and autocorrelation are the most important variables for prediction of the AIC. As Figure 3 showed, there were minimal differences between model performance but rather the flow period contributed most of the difference. This is reflected in the Random Forest models with the Period variable ranking highly. As better AIC scores are associated with drier conditions, the interpretation of the results here is that is lower values of Qcv and CF and higher AC result in better AIC scores.

### 4.4. Model Comparison

While not the main objective, this study does provide a comparison of model and objective function performance for a variety of catchments across Australia. In this case, we did not find any clear relationships between the number of parameters in the models and the performance. For most of the probability distribution curve (Figure 2) the simplest model GR4J achieves the greatest objective function scores. This echoes previous findings, where simpler models perform as good or better than parameter heavy conceptual models [7,9,70]. The most complex model, HBV, performed similarly to GR4J across the three different objective functions, showing that the complexity of this model is generally not warranted for the catchments modelled here. SIMHYD had the poorest performance, only achieving performance parity for the lower objective function values.

At this point, we have not analysed this difference in performance in more detail but this might be related to different conceptualisations of the catchment processes in the models. For example, the free parameter in the water exchange function in GR4J significantly improves low flow simulation [35,71] and the combination of the threshold activated and normal outflows in the upper storage layer in the HBV improve simulation for responsive catchments [36]. The differences in the linearity in the algorithms in models may cause differences in performance between perennial and ephemeral catchments.

### 4.5. Sources of Uncertainty

#### 4.5.1. Gap Filling

As acknowledged in the methods, the BoM gap fills the HRS data record using the GR4J model, which was also used as one of the study models. This may be a contributing factor in the GR4J model outperforming HBV and SIMHYD in both wet and dry periods. We minimised the effect of the gap filling by only using catchments that contained a greater than 90% quality code for the duration of the data record. For the final stations selected for use in this study, the data record for the wet and dry periods contained $96.60 \pm 3.54\%$ and $96.40 \pm 4.61\%$ of the highest quality code, respectively. Additionally, the proportions of the record that were gapfilled are $0.85 \pm 2.63\%$ and $1.05 \pm 3.22\%$ for the wet and dry periods. A study that examined the effect of gap filling with the same dataset as this study found that if the missing data rate is less than 10%, the gap filled dataset produced by a calibrated model was comparable to the benchmark data (less than 1% of the data missing) [72]. As we were interested in aggregate performance measures, we expect the impact of gap filling to be small on the results of this study.

#### 4.5.2. Observed Data

There is always uncertainty with input data used in hydrological models. Forcing data have limited spatio-temporal coverage and errors are created when interpolating to a grid [73,74] and observations of discharge are often inaccurate [75,76]. This makes the waterbalance difficult to close [77] and contributes to model uncertainty and the results of this study are no exception. We minimised the effect of uncertainty in discharge observations by constraining catchment selection to containing over 90% of the highest quality code. While there is uncertainty associated with the forcing data, there is little indication in Jeffrey et al. [42] that there is bias in the interpolation methods to a particular climatic period.

#### 4.5.3. Climate Classification

The classification of the dry period in this study is based on the Millennium Drought that occurred from 1999–2009 and was experienced in south eastern Australia [40]. This takes a more broad brush approach to dry and wet periods, rather than the more detailed approach suggested by Fowler et al. [8]. The final selection of catchments are located across Australia (Figure 1) and some are located outside the areas that may not have experienced the same dry conditions, notably in northern Australia and south western Australia. The PCA results were screened to check if the catchment location was the

major influence, rather than catchment type. The tropical stations showed no clear clustering or trend, however there is an apparent divide between east and west. The south western catchments are separated by the first principal component and placed with the ephemeral streams. This could be due to other differences, for example, Western Australia is geologically distinct from eastern Australia [78], is flat and has very sandy soil with low water holding capacity. In addition, since the early 1970s, south western Australia has experienced a prolonged decline in annual precipitation [79]. There were 10 catchments included from south western Australia and even if they are excluded, there was only a slight change in the results.

*4.6. Implications*

The implications of the results are that studies that apply the differential split sample test need to consider that the model itself might be biased towards one of the samples. In addition, there is a need to identify or develop a model that either performs better on erratic ephemeral systems or that can cover the whole spectrum of flow behaviours.

Finally, the analysis in this paper should be expanded to include more models, objective functions [34] and catchment characteristics [21] as this would be helpful to confirm the results obtained here. As with Euser et al. [55], we found the hydrological signatures to be highly correlated. This is reflected in the Random Forest results, where the collinear signatures are clustered with similar importance rankings. Collinearity is not an issue with the performance of Random Forest models, however, using additional signatures that are less correlated may be able to account for some of the differences observed between the models.

## 5. Conclusions

From the research it can be concluded that there appears to be bias in the calibration performance of the studied models (HBV, GR4J and SIMHYD) towards higher flows, perennial systems and steeper flow duration curves. This has implications for studies in calibration and validation of models (i.e., the differential split sample test) as well as for general water resource modelling applications. In addition to improve calibration methods, there is a need to develop a conceptual model that can deal with changes in climate and variation in catchments across the globe.

**Author Contributions:** This manuscript is the result of the initial Honours work of B.T. that was expanded substantially by A.J.V.B. Formal analysis, A.J.V.B.; Investigation, A.J.V.B.; Methodology, B.T., F.F.v.O. and R.W.V.; Supervision, R.W.V.; Writing—original draft, A.J.V.B. and R.W.V.; Writing—review & editing, A.J.V.B., F.F.v.O. and R.W.V.

**Funding:** This research received no external funding.

**Acknowledgments:** A.J.V.B. acknowledges the support of the Australian Government for the Research Training Program postgraduate award. The authors acknowledge the Sydney Informatics Hub and the University of Sydney's high performance computing cluster Artemis for providing the high performance computing resources that have contributed to the research results reported within this paper.

**Conflicts of Interest:** The authors declare no conflict of interest.

## Appendix A

**Table A1.** Model parameters, definition and range used in the analysis.

| Model | Description | Range |
|---|---|---|
| *GR4J* | | |
| x1 | Capacity of the production soil (SMA) store (mm) | 50–2000 |
| x2 | Water exchange coefficient (mm) | −10–10 |
| x3 | Capacity of the routing store (mm) | 5–500 |
| x4 | Time parameter (days) for unit hydrographs | 0.5–10 |
| *SIMHYD* | | |
| INSC | Interception store capacity (mm) | 0.5–5 |
| COEFF | Maximum infiltration loss (mm) | 50–500 |
| SQ | Infiltration loss exponent | 0–6 |
| SMSC | Soil moisture store capcaity (mm) | 50–2000 |
| SUB | Constant of proportionality in interflow equation | 0–1 |
| CRAK | Constant of proportionality in groundwater recharge equation | 0–1 |
| K | Baseflow linear recession parameter | 0.003–0.3 |
| *HBV* | | |
| FC | Maximum value of soil moisture storage (mm) | 50–2000 |
| $L_p$ | Fraction of FC above which AET equals PET | 0.3–1 |
| $\beta$ | Shape coefficient | 1–6 |
| $K_0$ | Recession coefficient | 0.05–0.5 |
| $K_1$ | Recession coefficient | 0.01–0.4 |
| $K_2$ | Recession coefficient | 0.001–0.15 |
| MAXBAS | Length of triangular weighting function in routing routine (days) | 1–7 |
| PERC | Maximum rate of recharge between upper and lower groundwater boxes | 0–3 |
| $U_{zl}$ | Threshold for quick runoff | 10–100 |
| CFMAX | Snow degree day factor (mm day$^{-1}$°C$^{-1}$) | 0–20 |
| $T_a$ | Snowmelt threshold (°C) | 1 |
| TT | Snowfall threshold (°C) | −1.419 |

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

**Sample Availability:** Model and analysis code is available on request.

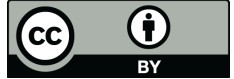

