# Peer review of "Conceptual Models and Calibration Performance—Investigating Catchment Bias"

_water, doi:10.3390/w11112424_

Round 1
Reviewer 1 Report
The manuscript number water-634077, entitled Conceptual Models and Calibration Performance: Investigating Catchment Bias, uses three different models (GR4J, HBV, and SIMHYD) and three objective functions to explore the relationship between model performance, catchment features, and identified parameters, under the hypothesis that the hydrological signatures and catchment characteristics reflect catchment behavior and that certain signatures and characteristics are associated with better calibration performance.
I found the theme of the manuscript very interesting and, after a careful reading, I think that the manuscript can be accepted for publication in the present form. The authors have done a very good job and are commended for their effort.
The authors can find a couple of suggestions in the attached pdf file.

Author Response
We thank the reviewer for the comments and feedback.
Addressing the comments related to Figure 4:
Reviewer comment 1: "I find the plot a little bit confusing. For example, not always I can couple an arrow to its variable.
Even though I understand it is not easy to do, authors should try to make it clearer."
We agree that this figure is somewhat unclear in the original manuscript. To address this we have: (1) made the figure cover a full landscape page and (2) coloured each of the vectors and the associated label to make it clearer which vector belongs to which variable.
There is still some crowding of the vectors labels, but the revised effort is the best we can achieve within the space requirements.
Reviewer comment 2: "I cannot understand what the authors mean."
This comment is in relation to the text in the manuscript: "The R2log objective function vectors are labelled as RSQL inside the biplots."
Inside each biplot is a vector representing the objective function that those model runs were calibrated on. Due to text formatting limitations of the plotting packages (ggplot2 and factoextra in R), we are unable to label the vector as R2log and instead originally labelled it as RSQL. As this was unclear, the text inside the biplots for R2log has been changed to RSquaredLog and accordingly the figure text has been updated to:
"The R2log objective function vectors are labelled as RSquaredLog inside the biplots."
Reviewer 2 Report
General comments
This manuscript investigates an interesting topic, namely, whether it is possible to identify specific catchment characteristics and hydrological signatures that result in better calibration performance for three popular conceptual rainfall-runoff models (GR4J, HBV and SIMHYD). The authors conclude: (a) there is a need to develop a conceptual model that can deal with changes in climate and variation in catchments across the globe, and (b) calibration methods have to be improved. The manuscript is written correctly and may be interesting for publication in Water. However, a few mistakes have to be corrected. Therefore, I recommend acceptance of the manuscript after minor revisions.
Specifically, the manuscript has to be written according to the instructions for Authors of the WATER Journal.
References have to be written in accordance with the Instructions for Authors of the journal. Journal articles (in References): Write the journal name in abbreviation form. For documents co-authored by a large number of persons (more than 10 authors) there are two possibilities, namely they can either cite all authors, or cite the first ten authors. In the present case of the 4 or 5 or 6 co-authors have to be cited all.Specific comments
Line 8
Please, define the abbreviations “GR4J”, “HBV” and “SIMHYD”.
NOTE: Abbreviations should be defined in parentheses the first time they appear in the abstract, main text, and in figure or table captions and used consistently thereafter Specifically, abbreviations should be defined in parentheses the first time they appear in the abstract, main text, and in figure or table captions and used consistently thereafter”.
Lines 13 & 14
The sentence “Models performed better in the wet period compared to the dry period, while GR4J generally fitted the data better” is unclear and it has to be written with explicitly.
Lines 14 & 15
Authors state: “There was little difference between the three objective functions”.
COMMENT: This sentence is vague. Please clarify, what does it mean the term “little difference”?
Lines 35 & 36
“….to other catchments (e.g. Figure 12 in Chiew et al. [17], Figure 6 in Coron et al. [18]).”
should be
“….to other catchments e.g. [17] (Figure 12), [18] (Figure 6).”
Line 64
“…performance (e.g. Figure 3 in Fowler et al. [31]). In other…”
should be
“…performance [31] (Figure 3]). In other…”
Line 66
“….Budyko and Dunne frameworks…” should be “….Budyko and Dunne approaches …”
Line 91
“For this study a range of catchments around Australia were identified….” Should be “For this study a range of catchments around Australia was identified…”
NOTE: The identification concerns the “range” not the “catchments”.
Line 105
“…basis of: 1. Streamflow…” should be “…basis of: (1) Streamflow…”
Line 106
“….periods, 2. Over 90% of….” should be “….periods and (2) over 90% of….”
Line 134
Please check. Is it correct “1-5”??
Equation (2)
Please. define: “Qsim” and “Qobs”.
Equations (2), (3), (5), (6)
Please, define “T”.
Lines 199 & 200, Lines 130-132
Authors state: “All of the catchment, model and objective function combinations successfully converged using the SCE-UA optimisation routine…”. However authors elsewhere (Lines 130-132) state: “Model optimisations were performed using the Shuffled Complex Evolution - University of Arizona (SCE-UA) algorithm [50]”. Consequently, the first sentence is incomplete. Please, check.
Line 207
“…for GR4J. SIMHYD had the…” should be “…for GR4J and SIMHYD had the…”
Figures 2 & 3
Check please: X-axis is %?
Line 538
“…Troch, P. A Decade of…” should be “…Troch, P. A. et al. Decade of…”
Author Response
We thank the reviewer for the feedback, helpful comments and suggested corrections to the manuscript.
We will address the specific comments individually:
Reviewer
Line 8
Please, define the abbreviations “GR4J”, “HBV” and “SIMHYD”.
Authors
Likely owing to their ubiquity, it is quite common that these models are not defined (e.g. https://www.mdpi.com/2073-4441/11/7/1328/htm, https://www.mdpi.com/2073-4441/11/10/2110, https://www.mdpi.com/2073-4441/10/10/1319/htm).
To partially accommodate the recommendation, GR4J and HBV are now defined the first time they are mentioned in the method. SIMHYD is not an abbreviation for anything. We consider defining the models once at their first mention appropriate given the above reasoning.
Reviewer
Lines 13 & 14
The sentence “Models performed better in the wet period compared to the dry period, while GR4J generally fitted the data better” is unclear and it has to be written with explicitly.
Authors
This section of the abstract has been amended to:
“The results show that a greater percentage of catchments achieved a better calibration performance in the wet period compared to the dry period and that better calibration performance is associated with catchments that have greater cumulative flow and a steeper flow duration curve.”
Reviewer
Lines 14 & 15
Authors state: “There was little difference between the three objective functions”.
COMMENT: This sentence is vague. Please clarify, what does it mean the term “little difference”?
Authors
This sentence has been removed as in the context of the paper it was to reinforce the performance bias to wetter conditions is not particular to a model/objective function pairing. An additional sentence has been added that states:
“The findings are consistent across the three models and three objective functions,
suggesting that there is a bias in the studied models to wetter catchments.”
Reviewer
Lines 35 & 36
“….to other catchments (e.g. Figure 12 in Chiew et al. [17], Figure 6 in Coron et al. [18]).”
should be
“….to other catchments e.g. [17] (Figure 12), [18] (Figure 6).”
Authors
This correction has been made.
Reviewer
Line 64
“…performance (e.g. Figure 3 in Fowler et al. [31]). In other…”
should be
“…performance [31] (Figure 3]). In other…”
Authors
This correction has been made.
Reviewer
Line 66
“….Budyko and Dunne frameworks…” should be “….Budyko and Dunne approaches …”
Authors
This correction has been made.
Reviewer
Line 91
“For this study a range of catchments around Australia were identified….” Should be “For this study a range of catchments around Australia was identified…”
NOTE: The identification concerns the “range” not the “catchments”.
Authors
This correction has been made.
Reviewer
Line 105
“…basis of: 1. Streamflow…” should be “…basis of: (1) Streamflow…”
Authors
This correction has been made.
Reviewer
Line 106
“….periods, 2. Over 90% of….” should be “….periods and (2) over 90% of….”
Authors
This correction has been made.
Reviewer
Line 134
Please check. Is it correct “1-5”??
Authors
This has been corrected to 10-5.
Reviewer
Equation (2)
Please. define: “Qsim” and “Qobs”.
Authors
Qsim and Qobs are now defined after the first appearance in the KGE objective function equation (Eq. 1).
Reviewer
Equations (2), (3), (5), (6)
Please, define “T”.
Authors
T is now defined after Eq. 2 and equations (2), (3), (5), and (5) have been amended to clarify that values at each timestep t=1 to t=T are evaluated.
Reviewer
Lines 199 & 200, Lines 130-132
Authors state: “All of the catchment, model and objective function combinations successfully converged using the SCE-UA optimisation routine…”. However authors elsewhere (Lines 130-132) state: “Model optimisations were performed using the Shuffled Complex Evolution - University of Arizona (SCE-UA) algorithm [50]”. Consequently, the first sentence is incomplete. Please, check.
Authors
I believe the confusion here is that the method sentence (lines 130-132) does not explicitly mention that SCE-UA is used for all objective functions and catchments. The sentence (lines 130-132) has been changed to: “Parameter optimisations were performed using the Shuffled Complex Evolution - University of Arizona (SCE-UA) algorithm [50].”
And an additional sentence has been ended at the end of the paragraph clarifying what is optimised:
“The SCE-UA optimiser is applied for each model, catchment and objective function combination, giving a total of 972 optimisations.”
Reviewer
Line 207
“…for GR4J. SIMHYD had the…” should be “…for GR4J and SIMHYD had the…”
Authors
This correction has been made.
Reviewer
Figures 2 & 3
Check please: X-axis is %?
Authors
The units (%) have now been specified in Figures 2 and 3.
e.g. the X-axis for Figure 2 now reads: Probability where objective function is exceeded (%).
Reviewer
Line 538
“…Troch, P. A Decade of…” should be “…Troch, P. A. et al. Decade of…”
Authors
This has been corrected. Instead of et al. after 10 authors, all authors have been cited instead (as was also suggested as an option by the reviewer in the general comments).